# RED : Looking for Redundancies for Data-Free Structured Compression of Deep Neural Networks

**Edouard Yvinec**[1,2] , **Arnaud Dapogny**[2] , **Matthieu Cord**[1] , **Kevin Bailly**[1,2]

Sorbonne Université[1], CNRS, ISIR, f-75005, 4 Place Jussieu 75005 Paris, France
Datakalab[2], 114 boulevard Malesherbes, 75017 Paris, France
`ey@datakalab.com`

## Abstract

Deep Neural Networks (DNNs) are ubiquitous in today's computer vision landscape, despite involving considerable computational costs. The mainstream approaches for runtime acceleration consist in pruning connections (*unstructured* pruning) or, better, filters (*structured* pruning), both often requiring data to retrain the model. In this paper, we present RED, a data-free structured, unified approach to tackle structured pruning. First, we propose a novel adaptive hashing of the scalar DNN weight distribution densities to increase the number of identical neurons represented by their weight vectors. Second, we prune the network by merging redundant neurons based on their relative similarities, as defined by their distance. Third, we propose a novel uneven depthwise separation technique to further prune convolutional layers. We demonstrate through a large variety of benchmarks that RED largely outperforms other data-free pruning methods, often reaching performance similar to unconstrained, data-driven methods.

## 1 Introduction

Modern Deep Neural Networks (DNNs) have become the mainstream approach in machine learning in general and in computer vision in particular, with CNNs achieving outstanding performance on various tasks such as object classification (He et al., 2016), detection (He et al., 2017) or segmentation (Chen et al., 2017). However, DNNs usually reach high requirements in terms of computational runtime. This prevents most state-of-the-art models to be deployed, most notably on edge devices. To address this shortcoming, a number of approaches for DNN compression have been proposed over the past few years. Architecture compression constitutes a convenient and popular way to address this runtime limitation, involving pruning as well as tensor decomposition techniques (Cheng et al., 2017). It consists in either removing connections, *i.e.* an *unstructured* way or suppressing or reordering specific channels or filters *i.e.* a *structured* fashion. Although the former usually removes more weights than the latter (Park et al., 2020), unstructured compression has the drawback to produce sparse weight matrices, which require dedicated hardware or libraries (Han et al., 2016) for real-case runtime improvements. Furthermore, these methods can also be divided in *data-driven* vs. *data-free* methods. While data-free methods are far more convenient for privacy concerns, as some data may be confidential (e.g. health data or military), they are still significantly outperformed by data-driven methods. Hence, despite recent work (Kim et al., 2020; Tanaka et al., 2020), data-free architecture compression remains a challenging and promising domain with room for improvements. In this paper, we propose RED , a novel data-free structured compression framework. First, RED leverages a novel adaptive scalar hashing of the layer-wise weight distributions to introduce redundancies in DNNs. In particular, we show that this hashing allows to introduce vector redundancies (*i.e.* neurons that perform the same operation) as well as tensor redundancies (*i.e.* low-rank flattened convolution kernels). These redundancies can respectively be exploited by applying similarity-based neuron

merging, as well as a novel uneven depthwise separation of convolutional layers. To sum it up, our contributions are:

- An adaptive scalar weight hashing technique based on local extrema search of the weight distribution density, that introduces redundancies among neurons without significantly altering the predictive function.
- A method for exploiting redundancies at the vector level with similarity-based neuron merging, and at the tensor level, with an uneven depthwise separation of convolutional layers that factors spatially redundant components.
- We introduce RED, a portable method for data-free structured DNN compression that significantly outperforms state-of-the-art data-free methods and often rivals existing data-driven approaches.

## 2 Related Work

Most architecture compression methods rely on an underlying approximation of the predictive function to later perform pruning, wether it can is *unstructured* or *structured* (as stated in Renda et al. (2020)), *data-driven* or *data-free*, *magnitude-based* or *similarity-based*.

**Predictive function approximation:** Perhaps one of the most studied such approximation is quantization (Nagel et al., 2019; Meller et al., 2019; Zhao et al., 2019). Quantization consists in mapping DNN weights to a finite, regular grid of values. It generally aims at reducing the inference time by coding this restricted set of values with fewer bytes (e.g. float16, int8 quantization), although it is generally non-adaptive to the weight distribution. Hashing constitutes another intuitive approach for predictive function approximation. Most hashing algorithms are extensions or variants of k-means (Lloyd, 1982) which requires a prior on the studied distribution to determine a fitting value for the number of clusters. This step can also be formulated as a learning problem (Wang et al., 2019; Stock et al., 2020), but requires data as a consequence. In this work we propose a data-free, prior-free and adaptive hashing of the scalar weight distribution, which introduces redundancies in DNNs.

**Structured Pruning:** On the one hand, unstructured approaches (Frankle & Carbin, 2018; Lin et al., 2020b; Park et al., 2020; Lee et al., 2020) consist in removing individual weights: hence, these methods rely on sparse matrices for implementation, which require dedicated hardware to fully exploit the reduction at inference time. On the other hand, the so-called structured approaches (Liebenwein et al., 2020; Li et al., 2017; He et al., 2018; Luo et al., 2017) aim at removing specific filters, channels or neurons. Although the latter usually results in less impressive raw pruning ratios as compared to the former, they allow significant runtime reduction without using dedicated hardware.

**Data-free Pruning:** Most pruning methods can be classified as *data-driven* as they involve, to some extent, the use of a training database. Lee et al. (2020) uses the drift of DNN weights from their initial values during training to select and replace irrelevant weights. The Hrank method (Lin et al., 2020a) consists in removing low-rank feature maps, based on the observation that the latter usually contain less relevant information. Lin et al. (2020b) extend the single layer magnitude-based weight pruning to a simultaneous multi-layer optimization, in order to better preserve the representation power during training. Other approaches, such as (Liebenwein et al., 2020; Meng et al., 2020), train an over-parameterized model and apply an absolute magnitude-based pruning scheme which removes a number of channels or neurons but generally causes accuracy drop. To address this problem, most of these methods usually fine-tune this pruned model for enhanced performance (Liu et al., 2018; Gale et al., 2019; Frankle & Carbin, 2018). Nevertheless, there exists a number of so-called *data-free* approaches which do not require any data or fine-tuning of the pruned network, however usually resulting in lower pruning ratios. For instance, Tanaka et al. (2020) is a data-free pruning method with lower performances but still addresses the layer-collapse issue (where all the weights in a layer are set to 0) by preserving the total synaptic saliency scores.

**Similarity-based Pruning:** All the aforementioned methods remove connections during training *via* more or less adaptive or learnable thresholds under which DNN weights are pruned. Hence, such paradigm constitutes an *absolute* pruning heuristic. However, *relative* methods based on comparison between neurons or feature maps have also been proposed in the literature. For instance, Ayinde

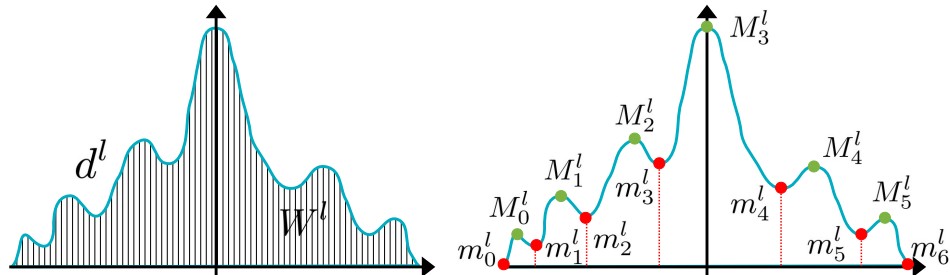

Figure 1: Illustration of the proposed adaptive scalar weight hashing. First (left), we find the estimate of the density function $d^l$ associated to the weights values $W^l$. Second (right), we find the local extrema $(m_k^l)_{k \in K_l^-}$ and $(M_k^l)_{k \in K_l^+}$ of $d^l$. Then we assign the new values $\tilde{W}^l \in K_l^{+n_l \times n_{l-1}}$.

et al. (2019) use a graph-based-group-average technique to define proximity between feature maps in order to prune filters. Recently, in Kim et al. (2020), similarity-based data-free pruning is applied by merging neurons. The authors decompose the weight tensor into new weights and a scaling matrix to fold into the next layer. However, in this approach, merging is performed at the tensor-level, rather than using a fine-grained pairwise similarity. Srinivas & Babu (2015), conversely, propose to merge together neurons of each layer based on pairwise vector similarity between them. However, their approach doesn't exploit scalar weight approximation, nor does it take into account tensor-level redundancies, resulting in low pruning ratios.

In this work, we introduce redundancies in DNNs by performing scalar hashing of the layer-wise weight distributions. In particular, we show that this hashing allows to introduce vector redundancies (*i.e.* neurons that perform the same operation) as well as tensor redundancies (*i.e.* low-rank flattened convolution kernels). These redundancies can respectively be exploited by applying similarity-based neuron merging in the same vein as Srinivas & Babu (2015), as well as a novel uneven depthwise separation of convolutional layers.

## 3 Introducing Redundancies via Adaptive Weights Hashing

Let's consider a DNN $f$ with $L$ layers $f = f_L \circ \cdots \circ f_1$. Each layer indexed by $l$ is defined by parameters $W^l$ for a $n_l$-dimensional output. As DNN weights take values in $\mathbb{R}$, the probability for two values to be equal is almost surely zero: as a consequence, the probability to have redundancies in DNNs (e.g. two neurons that perform the same operation, *i.e.* that have the same weight vector) is also zero, thus limiting the possible simplification of the predictive function $f$. To deal with this, we propose to simplify $f$ by first hashing the scalar weight distribution. As illustrated on Figure 1, we approximate the density function of the weights distribution for each layer $f_l$ using Kernel Density Estimation (KDE):

$$d^l : \omega \mapsto \frac{1}{n_l \times n_{l-1}\Delta_l} \sum_{w \in W^l} K\left(\frac{\omega - w}{\Delta_l}\right) \tag{1}$$

where $K$ is the density of a Gaussian kernel with bandwidth $\Delta_l$. Then, we estimate the local minima $K_l^- = (m_k^l)_k$ and maxima $K_l^+ = (M_k^l)_k$ of $d_l$ by computing its values over a discrete grid with range $[\min\{w \in W^l\}; \max\{w \in W^l\}]$. Because the KDE provides a continuous density function, the intermediate value theorem guarantees that $|K_l^+| + 1 = |K_l^-|$. We can thus partition $\mathbb{R}$ in $|K_l^+|$ intervals with boundaries defined by the local minima and assign the value of the local maximum to all parameters within the corresponding intervals. Assuming the $(m_k^l)_k$ and $(M_k^l)_k$ sorted, with $m_0^l = -\infty$ and $m_{|K_l^-|-1}^l = +\infty$, for every weight $w$ in $W^l$ there exists $k$ such that $w \in \left[m_k^l; m_{k+1}^l\right[$ and $\tilde{w} = M_k^l$. This defines the hashed layer $\tilde{f}^l$ with weights $\tilde{W}^l$. Note that this method is adaptive and has no prior on the weight distribution contrary to k-means.

In practice, we find that DNN weights concentrate around a limited number of local modes: hence, the proposed adaptive hashing dramatically reduces the number of different values that the weights can take, and introduces redundancies both at the vector and tensor level. Optionally, to conveniently enable further compression, we define (global and per-layer) contrast hyperparameters $\tau = \frac{1}{L}\sum_{l=1}^{L} \tau^l$.

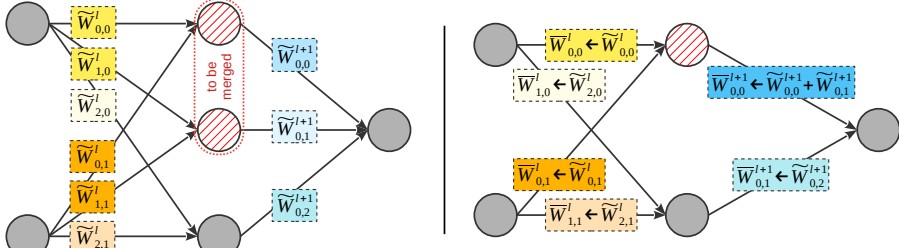

Figure 2: Neuron merging in the case of a fully-connected layer $l$ with weights $\tilde{W}^l_{i,j}$. Similar colors indicate equal weight values, e.g. $\tilde{W}^l_{0,0} = \tilde{W}^l_{1,0}$. The pruned network weights $\bar{W}$ are obtained by merging the first 2 neurons of layer $l$ and simply summing the corresponding weights in layer $l+1$.

These hyperparameters allow to define a distance threshold (relative to each layer's weight values range) under which two modes are collapsed to the dominant one, further increasing redundancies. We show in the experiments that hashing with $\tau = 0$ do not significantly alter the predictive function, thus setting $\tau > 0$ allows to find suitable trade-off between runtime acceleration and accuracy.

## 4 Exploiting Redundancies in DNNs

As the hashed weights $\tilde{W}^l$ take values in a finite set, redundancies are much more likely to occur both at the vector and tensor level.

**Vector Redundancies:** First, let's consider the case of a two-layers fully-connected neural network $f$ with an element-wise activation function $\sigma$ and no biases, *i.e.* $\tilde{f} : z \mapsto \tilde{W}^2 \sigma(\tilde{W}^1 z)$ with $\tilde{W}^1 \in K_1^{+ n_1 \times n_0}$ and $\tilde{W}^2 \in K_2^{+ n_2 \times n_1}$ the hashed parameters. Let's also assume that we have $\bar{n}_1 < n_1$ distinct neurons, *i.e.* $\bar{n}_1$ distinct rows in $\tilde{W}^1$. Let $\bar{W}^1$ be the sub-matrix of $\tilde{W}^1$ containing all the distinct rows of $\tilde{W}^1$ only once and $\bar{W}^2$ the matrix such that all columns from $\tilde{W}_2$ that were applied to identical neurons of $\tilde{W}^1$ are summed. Then, for each output dimension $i$ we have:

$$\left( \tilde{f}(z) \right)_i = \sum_j^{n_1} \tilde{W}^2_{i,j} \sigma \left( \sum_k^{n_1} \tilde{W}^1_{j,k} z_k \right) = \sum_j^{\bar{n}_1} \bar{W}^2_{i,j} \sigma \left( \sum_k^{\bar{n}_1} \bar{W}^1_{j,k} z_k \right) = \left( \bar{f}(z) \right)_i \qquad (2)$$

In other words, we can merge identical neurons (rows of $\tilde{W}^1$) and sum the corresponding weights in $\tilde{W}^2$ without altering the (hashed) layers outputs, as illustrated in Figure 2 on a simple case. This process can straightforwardly extended to neural networks with $L$ layers by repeating this process from the first to last layer. Furthermore, it can be adapted to:

- Layers with bias by considering $\tilde{W}' = (\tilde{W} \quad \tilde{b})$ and $z' = (z^T \quad 1)^T$
- Convolutional layers, by using a rewriting of the kernel, following Ma & Lu (2017)
- Batch-Normalization layers, by folding them like in Nagel et al. (2019)
- Skip Connections, where each output channel is computed using the corresponding weights in $\tilde{W}^l$ and $\tilde{W}^{l+k}$. Merge is performed on the two added layers simultaneously by considering the concatenation $\left( \tilde{W}^l \quad \tilde{W}^{l+k} \right)$.

In the case of depthwise convolutions (Sandler et al., 2018; Howard et al., 2019), each filter is applied to a distinct input. As a consequence, two filters intrinsically cannot perform identical operations and thus cannot be merged.

In practice, we find that after hashing, such exact neuron merging already allows to remove significant number of parameters. However, to obtain larger pruning ratios, this merging can be relaxed: for a layer $l$, we sort the set of pairwise distances between neurons or filters (e.g. as defined by the Euclidean distance between their weight vectors) and merge the $\alpha^l \%$ closest neurons by taking their average weight values. We thus define (global and layer-wise) hyperparameters $\alpha = \frac{1}{L} \sum_{l=1}^{L} \alpha^l$.

**Tensor Redundancies:** The proposed hashing method also induces tensor-level redundancies in regular 2D convolutional layers: to handle these redundancies we propose a novel uneven depthwise

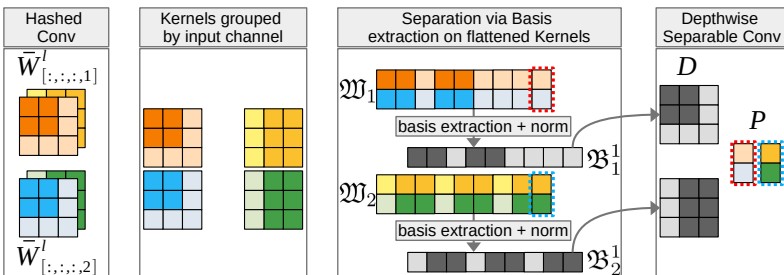

Figure 3: Tensor-level Simplification *via* of our uneven depthwise separation method for a layer with 2 input and 2 output channels. Restrictions of filters to each input channel $i$ are flattened and concatenated along the output channel axis, to form matrices $\mathfrak{W}_i$. If these matrices have rank $r_i$ we extract a basis $\mathfrak{B}_i^j$ (for $j$ in $[\![1; r_i]\!]$) of the rows of each $\mathfrak{W}_i$ and normalize it so that the last element is equal to 1. Then we deduce the weights of the pointwise kernels $P$ from the $(\mathfrak{B}_i^j)$ and the $\bar{W}$.

separation method. Let's consider a kernel $\bar{W} \in K^{+w \times h \times n_{\text{in}} \times n_{\text{out}}}$ for a layer $l$ after either the hashing or neuron merging steps (as merging and the proposed uneven depthwise separation steps can be applied in any order, see Appendix A.1). $\bar{W}$ can be expressed as a depthwise separable convolution if there exists $D \in \mathbb{R}^{w \times h \times n_{\text{in}} \times 1}$ and $P \in \mathbb{R}^{1 \times 1 \times n_{\text{in}} \times n_{\text{out}}}$ the respective weights of depthwise and pointwise convolution such that :

$$\forall y, x, i, j, \quad \bar{W}_{[y,x,i,j]} = D_{[y,x,i,1]} P_{[1,1,i,j]} \tag{3}$$

This condition can be expressed in terms of matrix ranks, as illustrated on Figure 3. We define the matrix $\mathfrak{W}_i$ as the restrictions of filters to input channel $i$, flattened and concatenated along the output channel axis:

$$\mathfrak{W}_i = \begin{pmatrix} \bar{W}_{[1,1,i,1]} & \cdots & \bar{W}_{[w,h,i,1]} \\ \vdots & \ddots & \vdots \\ \bar{W}_{[1,1,i,n_{\text{out}}]} & \cdots & \bar{W}_{[w,h,i,n_{\text{out}}]} \end{pmatrix} \tag{4}$$

Thus, any convolutional layer can be converted to a depthwise separable convolution layer if we have $r_i = 1$ for all $i$. In such case a basis $\mathfrak{B}_i^1 = D_{[:,:,i,1]}$ of the rows of the matrix $\mathfrak{W}_i$ can trivially be retrieved by considering the first non-zero row and the whole layer can be transformed to a depthwise convolutional layer. In practice, we find that it happens often after hashing. Otherwise, for all $i$ such that $r_i \neq 1$ we denote $(\mathfrak{B}_i^k)$ a basis of the rows of $\mathfrak{W}_i$, with $k \in [\![1; r_i]\!]$. Then, for every $j \in [\![1; n_{\text{out}}]\!]$ there exists $\mu_i^k \in \mathbb{R}$ and $k \in [\![1; r_y]\!]$ such that $\bar{W}_{[:,:,i,j]} = \mu_i^j \mathfrak{B}_i^k$. Thus, in this case, each output channel $j$ gives rise to a number of basis kernels $(\mathfrak{B}_i^k)_{k=1...r_i}$, and the corresponding subsequent point-wise convolutions with coefficients $\mu_i^j = P_{[1,1,i,j]}$, that depends of the rank of its concatenated flattened filter restriction matrices. Hence, we call this layer an uneven depthwise convolution with kernels defined as the basis kernels $\mathfrak{B}_i^k$.

To sum it up, the proposed RED method consists in three steps: an adaptive scalar hashing step (with hyperparameter $\tau$), followed by pruning *via* similarity-based neuron merging (with hyperparameter $\alpha$), and an uneven depthwise separation step (summarized as algorithms in Section A.2 of the supplementary material). As it will be shown in the experiments RED significantly outperforms other data-free architecture compression methods and often matches data-driven methods.

## 5 Experiments

### 5.1 Experimental setup

**Datasets and baselines:** we evaluate our models on the two *de facto* standard datasets for architecture compression, *i.e.* CIFAR-10 ((Krizhevsky et al., 2009), under the MIT License) and ImageNet ((Deng et al., 2009), under the BSD-3 license). We use the standard evaluation metric of removed parameters as well as removed FLOPs. We apply our approach on ResNet ((He et al., 2016), ResNet 20-56-110 and 164 with respective number of parameters 270k, 852k, 1.7M and 2.6M, and accuracies 92.48, 93.46, 93.81 and 94.54 on CIFAR-10 and ResNet 50 with 25M parameters and 76.17 accuracy on ImageNet) as well as Wide-ResNet (Zagoruyko & Komodakis, 2016). Wide ResNet architectures are defined by their number of layers as well as their wideness multiplier: we evaluate on Wide

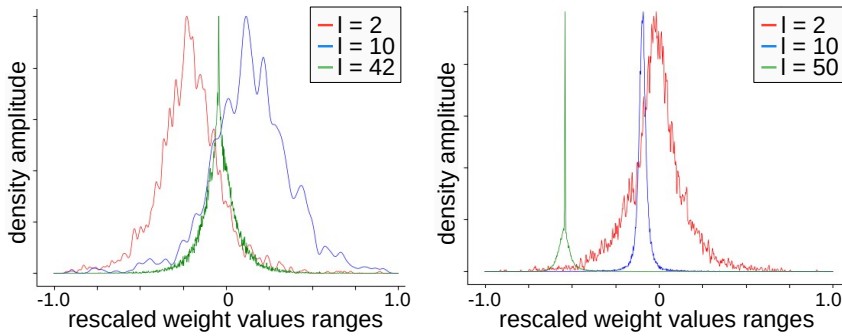

Figure 4: Weight distributions for several layers of a ResNet-56 on CIFAR-10 (left) and ImageNet (right) reveal that weights concentrate around multiple modes that can be captured *via* hashing.

ResNet 16-8 (with 11.0M and 95.2 accuracy on CIFAR-10), 22-2 (with 1.5 and 94.1 accuracy), 28 (28-2, 28-4, 28-8 and 28-10 with 1.9M, 7.4M, 29.8M and 36.5M parameters and 94.3, 94.8, 95.4 and 95.8 accuracies) and 40-4 (with 8.9M parameters and 95.0 accuracy).

**Implementation details:**   The proposed adaptive hashing is implemented using Scikit-learn python library, with bandwidth $\Delta_l$ set as the median of the differences between consecutive weight values per layer $l$ (see Appendix A.10) and the contrast hyperparameter $\tau$ set to 0, by default. Merging and depthwise separation are implemented using Numpy: for relaxed merging, and unless stated otherwise, hyperparameter $\alpha$ is set to the highest value that fully preserves the model accuracy (noted $\alpha^*$). We apply a per block strategy for setting the layer-wise $(\alpha^l)$ and a constant strategy for the $(\tau^l = \tau)$. Different strategy for setting the layer-wise $(\alpha^l)$ and $(\tau^l)$ are discussed in Section A.3 of the supplementary material. We ran our experiments on a Intel(R) Core(TM) i7-7820X CPU. The proposed hashing method processing time depends on the model's size: ranging from 45 to 413 seconds for ResNets on CIFAR-10 and up to 21 hours for a ResNet-50 on ImageNet. This step could however be accelerated by a layer-wise parallelization as discussed in Appendix A.9. Pruning, on the other hand, is much faster, taking 5 seconds for ResNet-20 on CIFAR-10 and 15 minutes for a ResNet-50 on ImageNet.

### 5.2   Hashing empirically induces vector and tensor-level redundancies

First, Figure 4 displays the weight distributions for several layers for ResNet-56 trained on CIFAR-10 and ResNet-50 on ImageNet. We observe that the weights concentrate around multiple dominant modes, with the number of modes and their relative proximity being variable from one layer to another: this motivates the design of the proposed adaptive hashing method, where each weight can be assigned to its corresponding mode, with optional collapse of adjacent modes.

In order to measure the impact of the change in the predictive function $f$ from the scalar hashing we considered the average error induced by hashing in logits $\mathbb{E}[|f(x) - \tilde{f}(x)|]$. This error is compared to the distribution of the differences between the top1 and top2 logits. An error smaller than the aforementioned difference implies that the modifications due to hashing can not change the prediction made by the DNN. For instance, we observe on ResNet 56 on CIFAR-10 an average error of 0.75 while the average difference between top 1 ad top 2 logits is $\approx 10.43$ in the baseline model with a standard deviation of 6.29. Furthermore, we observe similar values on other considered networks (for details see Appendix A.4), which empirically validates the efficiency of the hashing mechanism at preserving the accuracy.

Figure 5 also shows the evolution, during training and for each layer, of the removed parameter ratio (obtained with hashing and either merging, relaxed merging and RED , which consists in applying relaxed merging plus uneven depthwise separation). Initially, for the first epochs, there are no vector redundancies (left and central plots) and few tensor redundancies (right plot), due to the initialization scheme (e.g. Glorot or He) specifically designed to avoid vector redundancies. However, we observe a rapid convergence towards high numbers of both vector and tensor redundant weight distributions after only 10 epochs. In fact, we show in Section A.11 of the supplementary material that the effectiveness of RED is robust to the choice of the initialization scheme. Additionally, we show in Section A.5. of the supplementary material that we observe the same phenomenon even in the presence of training methods that aims at promoting diversity among neurons (e.g. dropout).

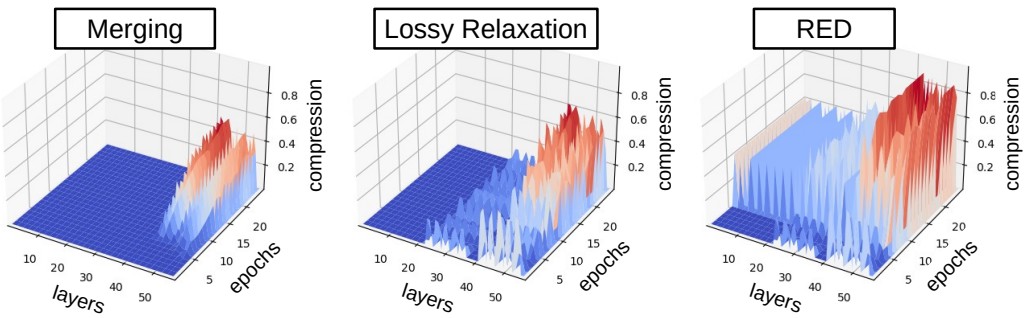

Figure 5: 3D plot of the removed parameters ratio as a function of the layer's depth and the training epoch for ResNet-56 on ResNet 50 CIFAR-10. Regardless of the initialization schemes designed to avoid redundancies, the hashed networks present large numbers of vector and tensor redundancies, that can be removed with minimal impact on the accuracy.

Table 1: Ablation results in terms of % removed parameters compared to the base model.

| Hashing | ✗ | ✗ | ✗ | ✓ | ✓ | ✓ |
|---|---|---|---|---|---|---|
| Merge | $\alpha = 0$ | $\alpha = \alpha^*$ | $\alpha = \alpha^*$ | $\alpha = 0$ | $\alpha = \alpha^*$ | $\alpha = \alpha^*$ |
| Depthwise Separation | ✗ | ✗ | ✓ | ✗ | ✗ | ✓ |
| ResNet 20 | 0.00 | 18.58 | 18.58 | 25.18 | 41.03 | **65.05** |
| ResNet 56 | 0.00 | 61.19 | 61.19 | 58.45 | 77.68 | **84.52** |
| ResNet 110 | 0.00 | 75.29 | 75.29 | 62.41 | 84.43 | **89.64** |
| ResNet 164 | 0.00 | 78.61 | 78.61 | 62.73 | 88.87 | **91.06** |
| Wide ResNet 16-8 | 0.00 | 31.08 | 31.08 | 19.67 | 38.67 | **51.92** |
| Wide ResNet 22-2 | 0.00 | 51.17 | 51.17 | 13.27 | 63.67 | **64.98** |
| Wide ResNet 28-2 | 0.00 | 49.19 | 49.19 | 11.46 | 61.20 | **64.19** |
| Wide ResNet 28-4 | 0.00 | 41.79 | 41.79 | 20.99 | 51.99 | **56.07** |
| Wide ResNet 28-8 | 0.00 | 33.58 | 33.58 | 19.78 | 41.78 | **52.87** |
| Wide ResNet 28-10 | 0.00 | 47.25 | 47.25 | 25.59 | 58.79 | **60.80** |
| Wide ResNet 40-4 | 0.00 | 49.67 | 49.67 | 43.37 | 61.80 | **70.35** |

As such, after hashing, merging exact redundancies removes $0 - 50\%$ of the weights, depending on the layer, with more emphasis on the last layers. Relaxed merging (which looks after similar but not necessarily equal neurons) removes up to $60\%$ of the layers' weights without changing the network accuracy. Furthermore, we can remove a lot of parameters ($40 - 85\%$) with uneven depthwise separation among all the layers.

## 5.3 Ablation study

Table 1 presents results in term of % removed parameters for each step in RED for both ResNet and Wide ResNet architectures on CIFAR-10. First, we observe that without hashing, the % removed parameters is significantly lower for every model and pruning method or combination thereof. For instance, on ResNet 20, Merge ($\alpha = 0$) with hashing (column 4) outperforms Merge ($\alpha = \alpha^*$) without hashing. Furthermore, it can be seen by comparing the first two columns that depthwise separation does not add much without hashing: therefore, we argue that scalar hashing is critical to introduce vector and tensor-level redundancies without accuracy loss.

As such, hashing + merging ($\alpha = 0$) already achieves good results without altering the predictive function (up to $62.7\%$ on deeper networks, e.g. ResNet-164). Furthermore, setting $\alpha = \alpha^*$ and using depthwise separation allows to reach higher removed parameters rates, e.g. up to $90\%$ on ResNet-110 and 164, without witnessing any accuracy drop. The experiments on Wide-ResNets show that the thinner the network the higher the removed parameters rates, with Wide ResNet 40-4 and 22-2 having the highest removed parameters ratio and 16-8 and 28-8 having the lowest. This is natural since, from a combinatorial standpoint, assuming a similar prior, the chance to find redundancies (strict or approximate) reduces as the channel depth increases. Last but not least, these results are very stable, as we observe standard deviations between $0.09 - 0.84$ for the different models over 10 runs

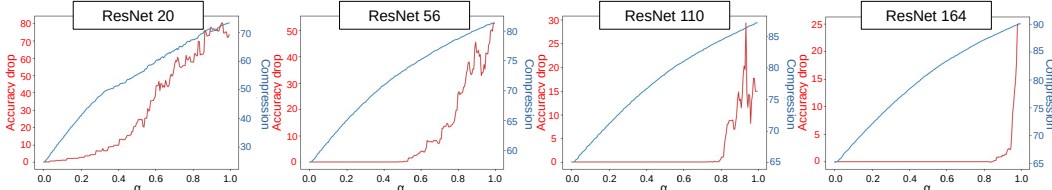

Figure 6: Accuracy drop (%, red) and compression (in term of % removed parameters, blue) vs. values of $\alpha$ for ResNet 20, 56, 110 and 164 on CIFAR-10. We can remove high number of similar neurons without impacting the accuracy, particularly for deeper networks.

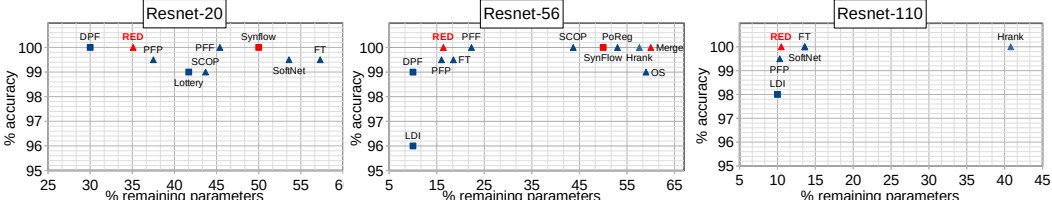

Figure 7: Comparison between RED and state-of-the-art methods on Cifar-10, in terms of % accuracy (measured as a percentage of the base model accuracy, the higher the better) and % remaining parameters (the lower the better). Each method is classified either as data-free (red) or data-driven (blue) and structured (triangle) or unstructured (rectangle). For each network, RED performs significantly better than other data-free methods, and often as well as data-driven or unstructured methods.

of retraining and application of RED.

Figure 6 shows the compression rate, and accuracy drop for relaxed merging as a function of parameter $\alpha$. For ResNet 20 we observe an initial low-slope phase for which merging neurons have a lower influence on the accuracy. For ResNet 56 and deeper, we observe an initial plateau where merging together a large number of similar neurons (up to 50% and 80% for ResNet 56 and 110-164, respectively) does not impact the accuracy at all. This further illustrates that the hashed DNN (and particularly deeper networks) weights present structural redundancies, and that similarity-based analysis can dramatically increase their efficiency. Note that the theoretical pruning factor of the merging and tensor decomposition steps are discussed in Section A.6 of the supplementary material. Also, we report the GPU and FLOPs performance of RED in Section A.7. Finally, hashing also affects memory footprint as evaluated in Appendix A.8.

## 5.4 Comparison with state-of-the-art approaches

**Comparisons on CIFAR-10:** Figure 7 draws a comparison between RED and state-of-the-art architecture compression methods, in terms of % remaining parameters and % accuracy w.r.t. the base model. For comparison purposes, we report results on the most popular architectures, *i.e.* ResNet 20, 56 and 110. Each of these approaches is either classified as:

- Data-driven (blue): methods that rely on fine-tuning or retraining protocols requiring labelled data, such as LDI (Lee et al., 2020), DPF (Lin et al., 2020b), PFP (Liebenwein et al., 2020), FT (Li et al., 2017), SoftNet (He et al., 2018), Lottery (Frankle & Carbin, 2018), PoReg (Zhuang et al., 2020), PFF (Meng et al., 2020), OS (Renda et al., 2020) and SCOP (Tang et al., 2020).
- Data-free (red): methods that do not use data such as Merge (Kim et al., 2020), SynFlow (Tanaka et al., 2020), or methods that generate synthetic data for fine-tuning such as Dream (Yin et al., 2020).

Another classification of these methods lies on the type of pruning: we distinguish unstructured methods (depicted by a square) from structured methods (triangle). Generally speaking, RED performs significantly better than all the others state-of-the-art data-free methods: it outperforms its closest contenders, SynFlow (Tanaka et al., 2020) on ResNet 20, as it removes 15% more parameters without accuracy drop. When compared to the most similar method (Kim et al., 2020), the merging step alone (with $\alpha = 0$) already achieves 18% higher removed parameters ratio on ResNet 56. Furthermore, the complete RED approach significantly widens the gap and reach 44% higher pruning ratio on ResNet 56. Furthermore, merging and RED also allows higher removed parameters ratios on Wide

Table 2: Comparison with state-of-the-art methods in term of % removed parameters/FLOPS and (top-1, top-5) accuracy on a ResNet-50 trained on ImageNet. Gray cells highlight data-free methods.

| Model | %removed parameters | %removed FLOPs | Top-1% | Top-5% |
|---|---|---|---|---|
| Baseline | - | - | 76.2 | 92.9 |
| GAL-0.5 | 16.9 | 43.0 | 72.0 | 90.9 |
| Dream$_{20}$ | 20.0 | 37.0 | 73.3 | - |
| SSS-32 | 27.1 | 31.1 | 74.2 | 91.9 |
| Hrank$_{1.58}$ | 36.7 | 43.7 | 75.0 | 92.3 |
| **RED** $_{\tau=0}$ | 39.6 | 42.7 | 76.1 | 92.9 |
| GAL-1 | 42.5 | 61.4 | 69.9 | 89.8 |
| SCOP | 42.8 | 45.3 | 76.0 | 92.8 |
| **RED** $_{\tau=0.05}$ | 42.1 | 44.5 | 75.3 | 92.1 |
| Hrank$_{1.85}$ | 46.0 | 62.1 | 72.0 | 91.0 |
| **RED** $_{\tau=0.10}$ | 47.3 | 47.9 | 74.1 | 91.1 |
| Dream$_{50}$ | 50.0 | 71.0 | 60.7 | - |
| **RED** $_{\tau=0.15}$ | 54.7 | 55.0 | 71.1 | 90.7 |
| **RED** $_{\tau=0.20}$ | 56.9 | 57.3 | 67.9 | 90.3 |
| Hrank$_{2.64}$ | 67.7 | 76.0 | 69.1 | 89.6 |

ResNet 40-4 (5.6% and 30%, respectively), due to finer pairwise similarity modeling as well as handling structural redundancies through uneven depthwise separation. RED also outperforms the best structured, data-driven method, PFP (Liebenwein et al., 2020) by achieving similar compression ratios on the three networks without any accuracy drop. Last but not least, RED, despite being a data-free approach, is competitive with recent data-driven and unstructured methods such as DPF (Lin et al., 2020b), Lottery (Frankle & Carbin, 2018) or LDI (Lee et al., 2020).

**Comparisons on ImageNet:** Table 2 shows a comparison between RED and other state-of-the-art structured pruning approaches on ImageNet. These methods allow to find different levels of compromise between the removed parameter ratio (as indicated by the % remaining parameters) and the accuracy drop.In order to find such trade-off with RED, we vary the contrast hyperparameter $\tau$ in the hashing step: by merging together close modes of the weight distribution, we introduce more redundancies, allowing to further compress the network at the expense of accuracy. Although this initial hashing step may cause the network accuracy to drop, both redundant neuron merging and uneven depthwise separation induce negligible loss, as echoed by the previous experiments. Thus, RED allows to remove between 39.6% (42.7% FLOPs) of a ResNet-50 with only 0.1% top-1 accuracy drop, and (with $\tau = 0.20$) 56.9% (57.3% FLOPs) with 8.3% top-1 accuracy drop. Thus, RED appears as a more efficient pruning algorithm than data-free Dream (Yin et al., 2020), allowing to remove large numbers of parameters with minimal accuracy drop. Furthermore, RED is once again very competitive with recent data-driven methods such as Hrank (Lin et al., 2020a), SCOP (Tang et al., 2020), SSS (Huang & Wang, 2018) or GAL (Lin et al., 2019). While some recent data-driven, unstructured approaches achieved higher levels of compression on this benchmark Evci et al. (2020), these results show the potential of RED as an efficient, portable and privacy compliant data-free, structured pruning method. In addition to that, we tested RED on more challenging backbones in Appendix A.12.

# 6 Conclusion

In this paper, we proposed RED , a novel data-free structured architecture compression method. First, RED uses a novel adaptive scalar hashing of the weight distributions to introduce redundancies in DNNs under the form of vector redundancies as well as tensor redundancies. These redundancies can be exploited with similarity-based neuron merging, as well as a novel uneven depthwise separation scheme for convolutional layers, respectively. Furthermore, we demonstrated through thorough experiments involving several architectures and databases, that RED significantly outperforms other data-free, structured pruning methods and often matches recent state-of-the-art data-driven pruning techniques, at a very minimal expanse in term of accuracy drop.

The whole approach is fully data-free and easy-to-use. On the one hand, DNN acceleration leads to either less energy consumption at inference or higher accuracy on a given device. On the other hand, the advantage of data-free methods is the compliance with data-privacy laws, particularly for applications where data access is sensible and the need to share such data may infringe privacy laws or make partnerships with external actors difficult.

It shall nevertheless be empathized that RED could theoretically be used in conjunction with other DNN compression techniques such as existing data-free quantization techniques. Furthermore, sparse pruning schemes, e.g. magnitude-based pruning methods could be considered to further reduce the computational runtime, given appropriate hardware. Last but not least, RED could quite straightforwardly be applied to any existing off-the-shelf computer vision model where runtime optimization is a concern, such as, for instance, object detection or semantic segmentation architectures.

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
