## A Appendix

### A.1 Commutativity

RED prunes a hashed neural network $\tilde{f}$ using two steps, a similarity-based pruning method (merge) and a tensor decomposition in an uneven depthwise separable convolution. These two steps are commutative. Let's consider a layer $l$ with hashed weights $\tilde{W}^l$. Output dimensions of $\tilde{W}^l$ are merged if and only if they are considered similar (either identical or within the $\alpha^l\%$ most similar. Let's note $i$ and $j$ the indices of two output dimensions that should be merged if merging was performed before the tensor decomposition. Now if we perform the tensor decomposition first, we get two tensors $D^l$ and $P^l$ such that

$$W_i^l = D^l \cdot P_i^l \tag{5}$$

$W_i^l$ are the weights corresponding to the $i^{\text{th}}$ output and $\cdot$ is the channel-wise product. As a direct consequence $\|W_i^l - W_j^l\|$ is among the $\alpha^l\%$ smallest distances if and only if $\|P_i^l - P_j^l\|$ is also among the $\alpha^l\%$ smallest distances as long as the $D^l$ are normalized. Thus, the pruning factor from merging is independent to the steps ordering. In the case of $\alpha = 0$ we have the same result for the uneven depthwise separable convolution. This is simple to see as the ranks in the uneven depthwise are computed per input and the merging is done by output.

### A.2 Algorithm

The proposed RED method is summarized in algorithm 1. Although our method is sequential, the two pruning steps can commute. The first step of RED is a data-free adaptive hashing step (Algorithm

---

**Algorithm 1** RED method

**Input:** trained DNN $f$, hyper-parameters $\alpha$ and $\tau$
$\tilde{f} \leftarrow$ Hashing $(f, \tau)$                                          ▶ Algorithm 2
$\bar{f} \leftarrow$ Merging $(\tilde{f}, \alpha)$                                          ▶ Algorithm 3
$\bar{f} \leftarrow$ Depthwise_Separation $(\bar{f})$                         ▶ Algorithm 4
return $\bar{f}$

---

2) which transforms the trained neural network $f$ in a hashed version $\tilde{f}$. As seen in Section 5.2, for certain layers, the KDE may have very close extremas, that can be fused depending of layer-wise hyperparameter $\tau^l$ (and global hyperparameter $\tau$), which defines the minimum contrast between two modes. However, note that setting $\tau^l = 0$ still allows very efficient hashing. The second step

---

**Algorithm 2** Hashing

**Input:** trained DNN $f$ with weights $(W^l)_{l \in [\![1;L]\!]}$, hyper-parameters $(\tau^l)_{l \in [\![1;L]\!]}$
Initialize $\tilde{f} = f$
**for** $l = 1$ **to** $L$ **do**
    $d^l = \text{KDE}(W^l)$
    extract $(m_k^l)_{k \in K^-}$ and $(M_k^l)_{k \in K^+}$ from $d^l$
    $(M_k^l)_{k \in K^+} \leftarrow \text{NMS}\left((M_k^l)_{k \in K^+}, \tau^l\right)$
    **for** $w \in W^l$ **do**
        find $k$ such that $w \in [m_k^l; m_{k+1}^l[$
        $\tilde{w} \leftarrow M_k^l$
    **end for**
**end for**
return $\tilde{f}$

---

(Algorithm 3) consists in a similarity-based merging of neurons where the similarity is computed as the euclidean distance between the weight values corresponding to each neurons. The process also adequately updates the consecutive layers. For the sake of simplicity we consider a sequential model without skip connections in this implementation. The layer-wise hyperparameter $\alpha^l$ (and global hyperparameter $\alpha$) defines the proportion of non-identical neurons to remove after ranking

**Algorithm 3** Merging Redundancies

**Input:** hashed DNN $\tilde{f}$, hyper-parameters $(\alpha^l)_{l\in[\![1;L]\!]}$
Initialize $\bar{f} = \tilde{f}$ with $(\bar{W}^l)_{l\in[\![1;L]\!]} \leftarrow (\tilde{W}^l)_{l\in[\![1;L]\!]}$
**for** $l = 1$ **to** $L - 1$ **do**
    $D \leftarrow$ matrix of $l^2$ distances between all neurons
    $d \leftarrow \alpha^l$ percentile of $D$                ▶ $d$ is the threshold distance
    $D_{i,j} \leftarrow \mathbb{1}_{D_{i,j}\geq d \text{ or } i=j}$         ▶ $D$ is a graph of neurons connected by similarity
    $M \leftarrow$ connected components from $D$
    $\bar{W}^l_{\text{new}} = [\,]$
    **for** comp $\in M$ **do**
        $\bar{W}^l_{\text{new}}.\text{append}\left(\frac{1}{|\text{comp}|}\sum_{j\in\text{comp}}\bar{W}^l_{[\dots,j]}\right)$       ▶ merge per connected component
    **end for**
    $\bar{W}^l \leftarrow \bar{W}^l_{\text{new}}$
    $\bar{W}^{l+1}_{\text{new}} = [\,]$                 ▶ We still have to update the layer $l + 1$
    **for** comp $\in M$ **do**
        $\bar{W}^{l+1}_{\text{new}}.\text{append}\left(\sum_{i\in\text{comp}}\bar{W}^{l+1}_{[i,\dots]}\right)$
    **end for**
    $\bar{W}^{l+1} \leftarrow \bar{W}^{l+1}_{\text{new}}$
**end for**
return $\bar{f}$

---

the pairwise distance between them. In particular, for $\alpha = 0$, we only merge identical neurons. The final step, only relevant to CNNs, (Algorithm 4), checks if convolutional layers can be converted in depthwise separable ones based on the criterion we introduced. Note that we didn't describe the situation where ranks are not all equal to 1 as this case only changes the depthwise implementation and basis extraction.

---

**Algorithm 4** Depthwise Separation

**Input:** merged DNN $\bar{f}$ with weights $(\bar{W}^l)_{l\in[\![1;L]\!]}$
**for** $\bar{f}^l$ convolutional layer of shape $w, h, n_{\text{in}}, n_{\text{out}}$ **do**
    **for** $i = 1$ **to** $n_{\text{in}}$ **do**
        $r_i \leftarrow \text{rank}\begin{pmatrix} \bar{W}_{[1,1,i,1]} & \cdots & \bar{W}_{[w,h,i,1]} \\ \vdots & \ddots & \vdots \\ \bar{W}_{[1,1,i,n_{\text{out}}]} & \cdots & \bar{W}_{[w,h,i,n_{\text{out}}]} \end{pmatrix}$
        $D_{[:,:,i,1]} \leftarrow \bar{W}_{[\dots,i,j]}$ for $j$ such that $W_{[\dots,i,j]} \neq 0$
        $P_{[1,1,i,j]} \leftarrow \bar{W}_{[x,y,i,j]}/D_{[x,y,i,j]}$ such that $D_{[x,y,i,j]} \neq 0$
    **end for**
    $f_d \leftarrow$ Depthwise conv layer of weight $D$
    $f_p \leftarrow$ Pointwise conv layer of weight $P$
    $\bar{f}^l \leftarrow f_p \circ f_d$
**end for**
return $\bar{f}$

---

### A.3 Hyper-parameters Application Strategy

In addition to a trained neural network, our method takes two hyper-parameters $\alpha$ and $\tau$ as inputs. The hyperparameter $\tau$ defines the average value of the per layers $\tau^l$ contrast hyperparameters of the adaptive hashing step. The modes within range $\tau^l$ of the total range of the distributions are collapsed to the maximum value among them. The hyperparameter $\alpha$ defines the average value of the per layers $\alpha^l$ proportion of non-identical neurons to merge for each layer. For each of these hyper-parameters we compare different strategies to allocate values to each individual $(\alpha^l)$ from $\alpha$ and $(\tau^l)$ from $\tau$. For $(\alpha^l)$ we tested the following strategies:

Table 3: Comparison between different strategies for $\alpha^l$ in terms of pruning factor for ResNet 56 on CIFAR-10, with constant $\tau = 0$.

| Strategy | % removed parameters |
|---|---|
| linear descending | 77.90 |
| constant | 78.69 |
| linear ascending | 80.35 |
| block | **84.52** |

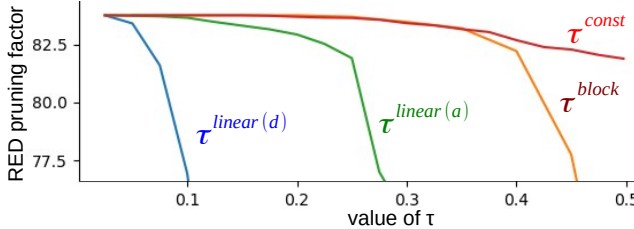

Figure 8: Evolution of the performance of RED in term of % removed parameters as a function of $\tau$ for the four tested strategies. The *constant* strategy provides the best results.

- per *block* strategy: we group $\alpha^l$ values per $1/3$ of the network layers such that:

$$\begin{cases} \alpha^l = \max\{2\alpha - 1, 0\} & \text{if } l \in [\![0; L/3[\![ \\ \alpha^l = \alpha & \text{if } l \in [\![L/3; 2L/3]\!] \\ \alpha^l = \min\{2\alpha, 1\} & \text{if } l \in ]\!]2L/3; L]\!] \end{cases} \quad (6)$$

- *constant* strategy: $\forall l \in [\![1; L]\!], \alpha^l$
- *linear ascending* strategy: $\alpha^l \ \forall l \in [\![1; L]\!], \alpha^l = \alpha l/L$
- *linear descending* strategy: $\forall l \in [\![1; L]\!], \alpha^l = \alpha(L-l)/L$

Table 3 draws a comparison between these different strategies in term of pruning ratio, with $\alpha$ set to the minimal value that does not bring any accuracy loss. We found the *linear descending* strategy, despite allowing to remove nearly $80\%$ of the network parameters, to be the least performing one, wollowed by the *constant* strategy. The *linear ascending* strategy is significantly better, validating the general idea that shallower layers contain more redundant information. Following this idea, the best performing strategy *block* allows to remove more than $4\%$ extra parameters: hence, we keep this per-block strategy in others experiments.

We evaluated the same strategies, *block*, *constant*, *linear ascending* and *linear descending* for the hashing contrast hyperparameters $(\tau^l)$. To evaluate each strategy for $\tau$ we measure the % of parameters removed with RED while keeping the accuracy constant: thus, the higher the pruning factor (with equal accuracy), the better the method. The results are showcased on Figure 8 for values of $\tau$ ranging from 0 to 0.5. We see that the *linear descending* and *linear ascending* strategies are the least performing, followed by the *block* strategy. The *constant* strategy is the best performing, thus we keep this constant setting in our experiments.

### A.4 Impact of Hashing

Following the study from Section 5.2, we want to empirically validate that hashing a DNN $f$ doesn't change the predictions. To do so we compare the error between the hashed DNN $\tilde{f}$ and the baseline model $f$ to the difference between the top1 and top2 logits in the original prediction from $f$. This is based on the fact that if the error is lower than the difference between the two logits with the highest responses then the highest logit will remain unchanged. We already provided values for ResNet 56 in the main paper. In Table 4, we provide values for more networks we benchmarked in pruning. As Stated in the main paper, the modifications from hashing are significantly smaller than the difference between the two highest logits of the baseline model. This is a beneficial consequence of the over-confidence of modern DNNs. This empirical observation validates the accuracy preservation after hashing and suggests that similar results could be achieved on other over-confident DNNs.

Table 4: Comparison between the modification induced by hashing (second column) and the confidence of the baseline model (third column), *i.e.* the difference between the highest and second highest logits.

| model | $\mathbb{E}[\|f(x) - \tilde{f}(x)\|]$ | $\mathbb{E}[\|\text{top}_1(f(x)) - \text{top}_2(f(x))\|]$ |
|---|---|---|
| ResNet 20 | 2.90 | 9.56 |
| ResNet 56 | 0.75 | 10.43 |
| ResNet 110 | 1.48 | 11.18 |
| ResNet 164 | 2.69 | 11.28 |
| Wide ResNet 28-10 | 1.24 | 10.95 |
| Wide ResNet 40-4 | 0.63 | 10.88 |

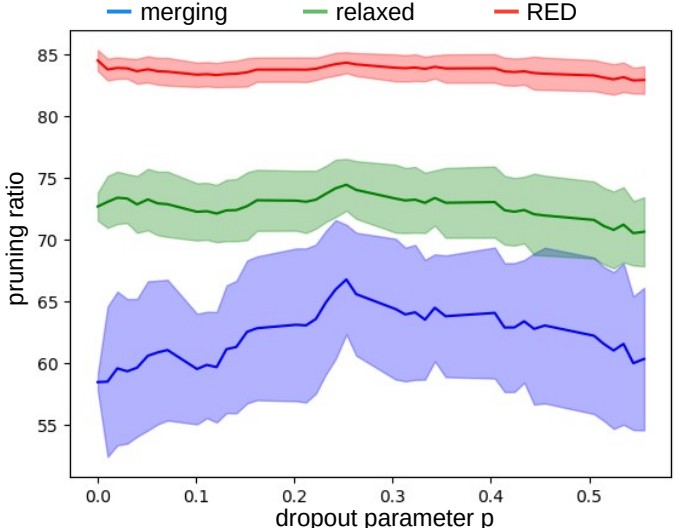

Figure 9: Graphs of the pruning factor resulting from the steps of RED as functions of the dropout parameter $p$. RED appears to be robust to dropout.

## A.5 Robustness to Dropout

Despite initialization methods used to avoid redundancies within the layers weights, other training or regularization methods exists with the purpose of exploiting available weights, thus potentially reducing redundancies, such as Dropout (Srivastava et al., 2014). Dropout is a vastly used deep learning technique that aims at avoiding overfitting by preventing co-adaptation between neurons. In practice, this is done by randomly dropping neurons at train time with a probability $p$. In order to evaluate the robustness of RED to different values of $p$ (modulating the intensity of the dropout), we retrained a ResNet-56 on CIFAR-10 with different values for $p$ on the last layer and then applied RED on these trained networks. Figure 9 shows the % removed parameters (average and standard deviation over 10 experiments) for each step of RED as a function of the dropout parameter $p$. First, surprisingly, we observe that Dropout causes an important rise of the average performance for the merging with $\alpha = 0$ (blue curve), with a significant variance. This is likely due to the fact the Dropout affects the last layers where most redundancies are found as seen in Figure 5 of the main paper. However, this effect is mitigated by the merging relaxation before completely vanishing after the uneven depthwise separation step. Furthermore we empirically observe that the ranks $r_i$ still converge to 1 in presence of Dropout. Based on these observations we can assess that, overall, the proposed method RED appears to be very robust to dropout.

## A.6 Expected Pruning Factor

The pruning factor can be estimated from the number of unique neurons per layer and the hyper-parameter $\alpha$. Let's consider a convolutional layer $l$ with shape $w \times h \times n_{\text{in}} \times n_{\text{out}}$ and an input $\alpha$ for the merging step and the corresponding $\alpha^l \in [0; 1[$ which indicates the proportion of unique neurons that shall be merged. Then, the pruned weight tensor will have a shape $w \times h \times n_{\text{in}} \times \lfloor \gamma^l (1 - \alpha^l) n_{\text{out}} \rfloor$,

Table 5: Percentage of FLOPs removed for different models on Cifar10.

| | % removed FLOPS | | | | | |
|---|---|---|---|---|---|---|
| Hashing | ✗ | ✗ | ✗ | ✓ | ✓ | ✓ |
| Merge ($\alpha = 0$) | ✓ | ✗ | ✓ | ✓ | ✓ | ✓ |
| Merge ($\alpha = \alpha^*$) | ✗ | ✓ | ✓ | ✗ | ✓ | ✓ |
| Depthwise Separation | ✗ | ✗ | ✗ | ✗ | ✗ | ✓ |
| ResNet 20 | 0 | 17.33 | 17.33 | 27.13 | 42.90 | **63.00** |
| ResNet 56 | 0 | 60.22 | 60.22 | 58.57 | 77.61 | **81.72** |
| ResNet 110 | 0 | 75.10 | 75.10 | 63.01 | 84.45 | **87.98** |
| ResNet 164 | 0 | 76.89 | 76.89 | 63.26 | 86.90 | **91.43** |
| Wide ResNet 16-8 | 0 | 30.78 | 30.78 | 20.07 | 39.20 | **52.04** |
| Wide ResNet 22-2 | 0 | 51.34 | 51.34 | 13.50 | 63.82 | **65.09** |
| Wide ResNet 28-2 | 0 | 50.87 | 50.87 | 11.16 | 60.91 | **64.10** |
| Wide ResNet 28-4 | 0 | 41.72 | 41.72 | 21.07 | 51.56 | **55.99** |
| Wide ResNet 28-8 | 0 | 33.28 | 33.28 | 19.80 | 41.62 | **53.17** |
| Wide ResNet 28-10 | 0 | 47.39 | 47.39 | 25.54 | 58.69 | **59.78** |
| Wide ResNet 40-4 | 0 | 49.51 | 49.51 | 43.41 | 61.99 | **70.09** |

Table 6: Runtime gain as a percentage of the removed inference time for different Cifar10 models on different hardware.

| device | batch size | ResNet 20 | ResNet 56 | ResNet 110 | ResNet 164 |
|---|---|---|---|---|---|
| RTX 3090 (GPU) | 256 | 62% | 75% | 85% | 89% |
| Intel m3 (CPU) | 32 | 87% | 88% | 88% | 89% |

where $\lfloor \cdot \rceil$ is the rounding operation and $\gamma^l$ is the proportion of unique neurons. The pruning ratio $r^l_{\text{merge}}$ at the end of this step is

$$r^l_{\text{merge}} = \gamma^l (1 - \alpha^l) \tag{7}$$

After the merging step we apply our depthwise separation technique to further compress the network, with the resulting pruning ratio $r^l_{\text{RED}}$ :

$$\begin{cases} r^l_{\text{RED}} = \frac{wh + r^l_{\text{merge}} n_{\text{out}}}{wh n_{\text{out}}} & \text{if } \forall i \in [\![1; n_i n]\!], r_i = 1 \\ r^l_{\text{RED}} = \frac{wh + r^l_{\text{merge}} n_{\text{out}}}{wh n_{\text{out}}} \frac{\sum_{i=1}^{n_{\text{in}}} r_i}{n_{\text{in}}} \end{cases} \tag{8}$$

where the $r_i$ are the ranks of the matrix obtained from the per input channel flattened weights, concatenated along the output channel.

## A.7  FLOPs and Inference-time

In Section 5.3 we evaluated RED using the standard metric of proportion of removed parameters. Another classic metric is the proportion of removed FLOPs which we provide here in Table 5. We observe that the results are similar to the values from Table 1. This is a consequence of the proposed pruning protocol which removes parameters in a structured way and removes more parameters on large convolutions (deep layers) as can be seen in Figure 5. Another important cause is the pruning of $3 \times 3$ convolutional layers which represent a large proportion of both parameters and FLOPs. Note that FLOPs removal are already provided for ImageNet in Table 2.

Another intuitive metric for pruning evaluation would be to compare runtime on CPU/GPU. However this metric presents many flaws, among which dependencies on the batch size, hardware and use of inference engines. Nonetheless we report the runtime gains over different hardware in Table 6. We measured the proportion of reduced computation time (i.e. the higher the better). We observe that on very small CPU (e.g. Intel m3) many operations are slowing the inference down, thus networks compression vastly impacts inference speed, although on ResNet 164 some operations remain bottleneck as the speed-up doesn't grow much with the pruning ratio. On the other hand, for large GPU (e.g. RTX 3090) we observe that the inference time reduction if strongly correlated to the pruning ratio and FLOPs removal. Overall RED enables a very effective speed-up of DNN inference from $60\%$ to $90\%$ while preserving the accuracy.

Table 7: Ablation results in terms of memory footprint reduction (ratio between zipped base and processed models).

| Model | zipped model memory ratio | | | |
|---|---|---|---|---|
| | hashing | $\alpha = 0$ | $\alpha = \alpha^*$ | RED |
| ResNet 20 | 12.36 | 12.86 | 12.95 | **21.69** |
| ResNet 56 | 23.29 | 25.71 | 27.76 | **41.34** |
| ResNet 110 | 35.74 | 38.45 | 43.04 | **58.33** |
| ResNet 164 | 25.40 | 25.40 | 52.92 | **66.84** |
| Wide ResNet 16-8 | 17.05 | 21.95 | 21.95 | **41.26** |
| Wide ResNet 22-2 | 12.07 | 12.93 | 12.98 | **30.17** |
| Wide ResNet 28-2 | 12.00 | 12.67 | 12.91 | **28.50** |
| Wide ResNet 28-4 | 15.15 | 17.88 | 17.88 | **33.11** |
| Wide ResNet 28-8 | 18.90 | 25.93 | 25.93 | **50.76** |
| Wide ResNet 28-10 | 20.32 | 29.34 | 29.34 | **47.00** |
| Wide ResNet 40-4 | 17.62 | 26.24 | 26.24 | **47.63** |

## A.8 Memory footprint Reduction

In this work we focus on pruning, nonetheless we also studied the consequence on the memory footprint of DNNs. To measure this impact we consider the ratio of the size of the processed networked zipped over the size of the original network also zipped. The empirical results are listed in Table 7. We observe that the hashing step alone has large influence on the memory footprint dividing it by 12 on already small networks (e.g. ResNet 20 and Wide ResNet 28-2) and up to 35 on larger networks (e.g. ResNet 110). The memory footprint is further reduced by pruning and tensor decomposition. Reaching 20 times reduction on ResNet 20 and up to 67 times on ResNet 164. Note that the zipping process aplies Huffman coding which depends on the distribution of the values to zip. For instance less values closer to a uniform distribution will less compressed that a larger list with a more peaky distribution. This is the case for some networks as their pruned version are not significantly smaller on disk once zipped compared to their hashed version.

## A.9 Scalability of the Hashing Algorithm

For a given layer $l$, the complexity of the proposed hashing is $\mathcal{O}(|W^l||S|)$ where $|W^l|$ denotes the number of weights and $|S|$ the sampling size (used for the evaluation of the kde and corresponds to $\omega$ in equation 1). As stated in the main paper hashing can be accelerated (e.g. up to 50 times faster on a ResNet 50) by processing the layers in parallel. Furthermore, for very large layers (e.g. layers with over $|W^l| = 10^6$ parameters) we can simply take a fraction (e.g. $5.10^4$ values in $W^l$) of the weights randomly to get identical density estimators much faster. Following the example, we would go $\frac{10^6}{5.10^4} = 20$ times faster, due to the linear complexity. As such, we were able to process ResNet 101 and 152 with $99.9\%$ compression and accuracy drop, in about 2 and 4.5 hours respectively.

## A.10 Choice of Bandwidth for Hashing

In the main paper, we computed the bandwidth parameter for the hashing step as the median difference of the (sorted) weight value set for each layer. The reasons behind this choice are two-fold. First, the bandwidth should depend on the range of the weight values which vary depending on the layer's depth and size. Second, it should be robust to outliers in the weight distribution. In order to validate this intuition, we consider the mean and median (as well as multiples of these values) as natural candidates for an adaptive value for the bandwidth. We report different set-ups in Table 8: the median provided the best trade-offs between accuracy preservation and compression rate, likely due to its robustness to outlier weight values.

## A.11 Robustness to Initialisation of RED

In order to assess the robustness of RED to different weight initialization schemes, we considered a ResNet-56 model on Cifar10. This model was initialized using either Xavier initialization Glorot & Bengio (2010) and He initialization He et al. (2015), using a similar training protocols apart from

Table 8: Hashing performance on ResNet 56 on Cifar10 for different strategies for setting $\Delta_l$. The compression is measured as the percentage of removed distinct weight values.

|  | base model | $\frac{median}{10}$ | median | 10×median | $\frac{mean}{10}$ | mean | 10×mean |
|---|---|---|---|---|---|---|---|
| accuracy | 93.46 | 93.46 | 93.41 | 92.95 | 93.46 | 93.40 | 93.02 |
| compression | 0% | 94.0% | 99.0% | 99.3% | 92.8% | 97.6% | 99.0% |

the initialization scheme. Our results are reported in Table 9 which empirically demonstrate the robustness of the proposed method to the initialisation scheme.

Table 9: Comparison of the influence of the initialisation scheme on the proposed method for a ResNet 56 trained on Cifar10.

|  | original acc | hashed acc | hashed and pruned acc | hashing ratio | pruning ratio |
|---|---|---|---|---|---|
| Xavier [1] | 93.64 | 93.52 | 93.52 | 99.0% | 84.47% |
| He [2] | 93.46 | 93.41 | 93.41 | 99.0% | 84.52% |

## A.12 Pruning Smaller Backbones

In this section, we consider a more challenging architecture for pruning, *i.e.* a smaller popular backbone on ImageNet, MobileNet v2 Sandler et al. (2018) with width multiplier 0.35. In particular, RED was able to remove $19.1\%$ parameters with no accuracy drop (*i.e.* using $\tau = 0$) on this challenging setup. Overall, RED achieves 20% to 30% pruning ratio by varying the contrast hyperparameter $\tau$.

Table 10: RED performance on a MobileNet V2 backbone (width multiplier 0.35) on ImageNet. We report the accuracy as well as the pruning ratio for different values of $\tau$.

|  | original model | $\tau$=0% | $\tau$=10% | $\tau$=20% | $\tau$=30% |
|---|---|---|---|---|---|
| accuracy | 60.3 | 60.3 | 56.9 | 46.7 | 45.0 |
| compression | 0% | 19.1% | 20.9% | 29.4%. | 30.0% |