# OpenReview forum: "RED : Looking for Redundancies for Data-FreeStructured Compression of Deep Neural Networks"
_NeurIPS.cc/2021/Conference — NeurIPS 2021 Poster_

### Official Review · Reviewer_tP4V · 2021-07-15

**Rating:** 5
**Confidence:** 3

**Summary:**

The author propose a data free pruning pipeline to compress a given deep neural network. The author propose to first apply adaptive weight hashing to force the weights tensors to be similar and then merge weight tensors based on a calculated similarity score. The proposed methods use kernel density function for hashing and do not require training data for hashing the weights tensors. The proposed pipeline (RED) is verified on CIFAR-10 and ImageNet.

**Ethics Review Area:**

["I don’t know"]

**Limitations And Societal Impact:**

The author do not provide social impact of the paper

**Main Review:**

Strength
1. The proposed pipeline (Hashing + Merging) is interesting and the author has demonstrated the effectiveness of hashing on introducing redundancies on the weights tensor.

2. Data-free compression is also worth to explore for its efficiency on computation resources.

Weakness
1. The writing of the paper need further refinement. For an example, table 1 in the paper seems to be incomplete as there is a missing of the column headers. I cannot figure out the meaning of those results based on the current version.

2. The impact of hashing on the model accuracy is not clear. While the hashing introduce tensor level redundancies, will it hurts the model representation capability?

3. How is the proposed methods affected to different initialization methods? This is mentioned in the caption of Figure 5 but the impacts are not clear as there is not experiment results on this point. I thought this is worth to explore.

4. The proposed method is mainly verified on large models like ResNet. It is not clear if the proposed method will work on more efficient backbones such as MobileNet [1], EfficientNet [2] and MobileNeXt [3].

Overall, the proposed method is interesting to explore. However, the current presentation and ablation experiments do not convince me on the effectiveness of the proposed RED pipeline. Besides, the difference between the proposed hashing methods and the quantization algorithm (as there are a lot of well performing quantization algorithms) is also worth to explore.

I am willing to incease my score if the author could address mu concerns sufficiently.

[1] Sandler, Mark, et al. "Mobilenetv2: Inverted residuals and linear bottlenecks." Proceedings of the IEEE conference on computer vision and pattern recognition. 2018.

[2] Tan, Mingxing, and Quoc Le. "Efficientnet: Rethinking model scaling for convolutional neural networks." International Conference on Machine Learning. PMLR, 2019.

[3] Zhou, Daquan, et al. "Rethinking bottleneck structure for efficient mobile network design." Computer Vision–ECCV 2020: 16th European Conference, Glasgow, UK, August 23–28, 2020, Proceedings, Part III 16. Springer International Publishing, 2020.


**Time Spent Reviewing:**

8

---

> ### Author Response · Authors · 2021-08-09
> **Authors' response**
>
> We thank the reviewer for their review and valuable feedback. Please find our answers below:
>
> **1-Table 1 headers:** Each column corresponds to the set up indicated by the marks \cmark and \xmark. For instance the first column of results corresponds to the result applying no hashing, merging with $\alpha =0$ and no depthwise separation, while the last one corresponds to using hashing, merging with $\alpha=\alpha^*$ and depthwise separation.
>
> **2-Impact of hashing:** we discussed the impact of hashing on the accuracy l.215-219. In table 1, the accuracy for all models is fully preserved (which is also the case for the pruned DNNs with $\alpha=0$, as showed in e.g. Figure 6 and 7): the accuracy of the hashed model is equal to the accuracy of the original model. Therefore, hashing does not hurt the representation capacity of the models. This is further detailed in Appendix A.4 where it can be seen that, for each model, the difference between the hashed and original model logits $\mathbb{E}[|f(x) - \tilde f(x)|]$ is much less than the difference between the top1 and top2 logits $\mathbb{E}[|\text{top}_1(f(x)) - \text{top}_2(f(x))|]$.
>
> **3-Effect of initialization:** for both Glorot and He initialization schemes, we observe that the weight values distributions after training are similar, thus the hashing is robust to initialization protocol. For instance, the ResNet and Wide-ResNets on Cifar-10 were trained with different initialization schemes. We mention these schemes to put the emphasis on the fact that initialized but untrained networks are harder to hash due to the distributions used, whereas trained networks can be hashed and pruned efficiently. We will make this point clearer in the final version of the paper.
>
> **4-Efficiency on smaller backbones:** to answer your concern, we conducted preliminary experiments on the smallest MobileNet v2 architecture with width multiplier 0.35 on ImageNet. RED removed between $20$% and $30$% parameters (for different values of $\tau$). We will add these results to the final paper, should it be accepted for publication.
>
> **Motivation of the use of hashing over quantization:** we also compared with the int8 quantization method from [1] on MobileNet v2 and observed that int8 quantization induces a drop of $3.96$% accuracy while our hashing method only does not cause any significant accuracy drop, for an equivalent number of removed weight values. We believe that is due to the fact that such uniform quantization method is intrinsically more constrained, whereas the proposed hashing more efficiently adapts to the weight distributions. We hope that these clarifications answer your concerns in a satisfying way.
>
> **Societal Impact:** On the one hand, DNN acceleration leads to either less energy consumption at inference or higher accuracy on a given device. On the other hand, the advantage of data-free methods is the compliance with data-privacy laws, particularly for applications where data access is sensible and the need to share such data may infringe privacy laws or make partnerships with external actors difficult.
>
> [1] Raghuraman  Krishnamoorthi. "Quantizing  deep  convolutional networks  for  efficient  inference:  A  whitepaper." arXiv  preprintarXiv:1806.08342, 2018.

---

> > ### Comment · Reviewer_tP4V · 2021-08-26
> > **The main concerns are remained**
> >
> > Thank you for your response. However, my main concerns on the impacts on the efficient backbones and the robustness of the proposed method to initialization remained.
> >
> > For example, authors claim that RED can be applied on MBV2 with 20% - 30% parameters removed. What about the accuracy drop? why select MBV2 with 0.35 as width multiplier? Similarly, the details of the experiments on the robustness to different initialization strategy are missing.
> >
> > Thus, I keep my initial rating.

---

> > > ### Author Response · Authors · 2021-08-27
> > > **Authors' response**
> > >
> > > **Efficiency on smaller backbones:** From your concerns about the efficiency of the method on more efficient backbones, we took what would appear to be the most challenging architecture, *i.e.* the smallest popular backbone on ImageNet, i.e. MobileNet v2 with width multiplier 0.35.
> > > On this challenging setup we achieve 20\% to 30\% pruning ratio, as showcased on the following Table. In particular, RED was able to remove $19.1$\% parameters with no accuracy drop (*i.e.* using $\tau=0$).
> > >
> > > |            | original model | $\tau$=0\%  | $\tau$=10\% | $\tau$=20\% | $\tau$=30\% |
> > > | --- | --- | --- | --- | --- | --- |
> > > | accuracy    | 60.3           | 60.3        | 56.9        | 46.7        | 45.0 |
> > > | compression | 0\%            | 19.1\%      | 20.9\%      | 29.4\%.     | 30.0\% |
> > >
> > >
> > > **Effect of initialization:** As stated in our first rebuttal, the models considered in Cifar10, namely Wide-ResNets and ResNets, were trained with different initialization schemes, namely Xavier initialization [1] and He initialization [2] respectively. To specifically answer your concern about initialization, we conducted an experiment with two ResNets 56 on Cifar10, one initialized using Xavier initialization [1] and the other with He initialization [2]. The training protocol were identical in both cases except for the initialization scheme. Our results are reported in the following Table:
> > >
> > >
> > >
> > > |            | original acc | hashed acc | hashed and pruned acc | hashing ratio | pruning ratio |
> > > | --- | --- | --- | --- | --- | --- |
> > > | Xavier [1] | 93.64        | 93.52      | 93.52                 | 99.0\%        | 84.47\% |
> > > | He [2]     | 93.46        | 93.41      | 93.41                 | 99.0\%        | 84.52\% |
> > >
> > > This confirms that RED is robust to different network initialization strategies. We hope that this answers your concerns in a satisfying way.
> > >
> > >
> > > [1] Glorot, Xavier and Bengio, Yoshua "Understanding the difficulty of training deep feedforward neural networks" JMLR 2010
> > >
> > > [2] He, Kaiming and Zhang, Xiangyu and Ren, Shaoqing and Sun, Jian "Delving deep into rectifiers: Surpassing human-level performance on imagenet classification" ICCV 2015

---

### Official Review · Reviewer_xhzB · 2021-07-16

**Rating:** 7
**Confidence:** 4

**Summary:**

This paper introduce a data-free compression methods by combing three strategies: adaptive scalar hashing, merging similar neurons and uneven depth-wise separation method to prune conv layers, obtaining SOTA data-free compression performance.

**Limitations And Societal Impact:**

Please provide Societal Impact discussion.

**Main Review:**

1. Novelty is pretty good. The observations of redundancies in DNN are pretty novel.
2. The comparisons conducted on Imagenet are not very strong. There are many published data-driven compression papers. The listed paper's performances are not real SOTA. However, this method is data-free, I will buy it.  It's better to show more comparisons on other benchmark models (for example, MobilenetV2) on Imagenet.
3. As mentioned in paper, this method can incorporate with other compression methods together for further compressions.

I am curious about:  What if we apply three strategies in loop iteratively? Can it improve the performance?

**Time Spent Reviewing:**

2 hours

---

> ### Author Response · Authors · 2021-08-09
> **Authors' response**
>
> We thank the reviewer for their review and insightful questions. Please find our answers below:
>
> **Comparisons on ImageNet are not very strong?** As you pointed out, the proposed method is data-free and structured: hence, we mainly compared RED with other structured pruning methods on ImageNet, with a special emphasis on data-free methods such as Dream [1]. However, it is true that these methods are generally outperformed by unstructured and data-driven ones. We will clarify this in the final version of the paper.
> To answer your concern about additional architectures, we conducted preliminary experiments on MobileNet v2 (the smallest backbone, with width multiplier 0.35) on ImageNet: the proposed approach removed between $20$% and $30$% parameters (for different values of $\tau$). We will add these experiments to the final version, shall it be accepted for publication.
>
> **Iteratively apply the proposed method?** Note that the depthwise separation can only be performed once (after hashing). Thus, the question becomes: can we iterate over the hashing and merging steps? This is an interesting question, our intuition suggests that this could lead to a little more compression and pruning (e.g. by hashing the new weight distribution corresponding to layers with merged neurons).
>
> **Societal Impact:** On the one hand, DNN acceleration leads to either less energy consumption at inference or higher accuracy on a given device. On the other hand, the advantage of data-free methods is the compliance with data-privacy laws, particularly for applications where data access is sensible and the need to share such data may infringe privacy laws or make partnerships with external actors difficult.
>
> [1] Yin, H., Molchanov, P., et al. "Dreaming to distill: Data-free knowledge transfer via deepinversion." CVPR 2020

---

> > ### Comment · Reviewer_xhzB · 2021-09-01
> > **Thank you for your response**
> >
> > After reading the responses to all reviews and the provided additional experiments results, I still keep my rating the same as before.  Thanks!

---

### Official Review · Reviewer_HdEN · 2021-07-21

**Rating:** 6
**Confidence:** 3

**Summary:**

### Summary
This paper proposes a three-step compression of neural networks without fine-tuning, which comprises weight hashing, merging of similar weights, and depthwise separation via basis extraction. The compression is filter-based, data-free, and with minimal accuracy degradation.


**Limitations And Societal Impact:**

Sufficiently discussed in the paper.

**Main Review:**

## Strengths
The combined approach appears to be effective and competitive against related works which includes some recent fine-grained and data-driven methods.

The reviewer believes that the method proposed is interesting with the use of density estimation for redundancy discovering, despite some presentations issues (as discussed below).

The experiments and ablation analyses appear to be sufficient to justify the claims in the paper.

## Weaknesses
* It is not immediately clear why hashing weights helps with identifying redundancy. Why not simply merge neurons with small $\ell^2$ distances? Perhaps providing a formal definition of how $\tau$ is used would clear up the confusion.
* It is also not clear how the KDE bandwidth $\Delta_l$ is determined. The hashing process is highly dependent on this.
* The proposed hashing method may have a scalability problem (“ranging from 45 to 413 seconds for ResNets on CIFAR-10 and up to 21 hours for a ResNet-50 on ImageNet”, line 202). It appears to not extrapolate well to larger models. What about ResNet-$N$ ($N \geq 101$)? Could you explain why this is the case or provide a discussion on its complexity?
* Minor issues:
	* Line 112 discreet -> discrete
	* Line 123 defines the contrast hyperparameter $\tau$, but provides no definition of how it is used.
	* Line 109: grammar: “for a each layer”.
* The combined method appear to be overall too complex, the reviewer would appreciate if the work will be open-source in the future.


**Time Spent Reviewing:**

5 hours

---

> ### Author Response · Authors · 2021-08-09
> **Authors' response**
>
> Thank you for your valuable feedback. We hope that the following clarifications answer your concerns and will allow to enhance the paper quality for the final version.
>
> **Why hashing helps with identifying redundancies and formal definition of $\tau$?** Columns 2 and 3 of Table 1 show the % of removed parameters obtained using merge (with $\alpha=0$ and $\alpha=\alpha^*$, respectively, and $\tau=0$ in both cases) directly without hashing. While trivially with $\alpha=0$ the chances to have exact redundancies is almost zero (as discussed in Section 3 l.103-110), with $\alpha=\alpha^*$ the results are significantly better with hashing (Table 3 col. 5-6). This is due to a more efficient scalar weight assignment using the proposed adaptive hashing. $\tau$ defines the minimal distance (normalized by the support of the weight distribution, to ensure $\tau \in [0,1]$) between two kept extrema during hashing. It is exploited in the experiments on ImageNet (see Table 2) to find different trade-offs between accuracy and pruning ratios.
>
> **Choice of bandwidth $\Delta_l$:** The bandwidth $\Delta_l$ is defined as follows: assuming $(\omega_i)$ is a sorted list of the scalar weight values of $W^l$ then $\Delta^l$ is set as the median of the set {$\omega_{i+1}-\omega_i$} (see l.194). The reasons behind this choice are two-fold. First, the bandwidth should depend on the range of the weight values which vary depending on the layer's depth and size. Second, it should be robust to outliers in the weight distribution.
>
> **Scalability of the hashing algorithm (e.g. ResNet 101)?** For a given layer $l$, the complexity of the proposed hashing is $\mathcal{O}(|W^l||S|)$ where $|W^l|$ denotes the number of weights and $|S|$ the sampling size (used for the evaluation of the kde and corresponds to $\omega$ in equation 1). As stated in l.202-203 hashing can be accelerated (e.g. up to 50 times faster on a ResNet 50) by processing the layers in parallel. Furthermore, for very large layers (e.g. layers with over $|W^l| = 10^6$ parameters) we can simply take a fraction (e.g. $5.10^4$ values in $W^l$) of the weights randomly to get identical density estimators much faster. Following the example, we would go $\frac{10^6}{5.10^4} = 20$ times faster, due to the linear complexity. As such, we were able to process ResNet 101 and 152 with $99.9$% compression and accuracy drop, in about 2 and 4.5 hours respectively.
>
> **Minor Issues:** The typos have been corrected and the contrast parameter's definition from line 123 will be extended in the future version of the article.

---

> > ### Comment · Reviewer_HdEN · 2021-08-26
> > **Thanks for the rebuttal, my concerns remain.**
> >
> > > Why hashing helps and the definition of $\tau$.
> >
> > I believe the paper in its current form
> > is still missing many important details
> > regarding the implementation of your method.
> > The hashing method partitions the weight distribution
> > into $\vert K^+_l \vert$ intervals.
> > However, it is still not clear how $K^+_l$
> > relates to the hyperparameters $\tau$ chosen
> > in your experiments.
> > To confuse things further,
> > in your reply,
> > you specified that $\tau \in [0, 1]$,
> > however in the experimental results in Table 2,
> > $\tau$ equals 5, 10 or 20.
> > Please provide a formal definition of $\tau$.
> >
> > > Choice of bandwidth
> >
> > I believe you should justify the choice of the value
> > with additional experiments.
> >
> > > Scalability of the hashing algorithm
> >
> > It is unclear if there are accuracy implications
> > of the acceleration strategies,
> > and they should also be validated with experiments.
> >
> > Overall,
> > the rebuttal did not clarify the problems
> > related to the method / presentation,
> > and did not present evidence
> > regarding the design decisions.
> > The reviewer maintains the current rating.

---

> > > ### Author Response · Authors · 2021-08-27
> > > **Authors' response**
> > >
> > > **Formal definition of $\tau$:** our contrast parameter $\tau \in [0;1]$ (in the article the values 5,10,.. are percentages and we will fix this error) defines the minimal normalized distance between two kept extrema as previously stated. In other words, assume our kde-based hashing protocol gave the $(M_0^l,...,M_n^l)$ maxima for a layer $l$. For each such maxima we have a probability (given by the kde) $(d^l(M_0^l),..,d^l(M_n^l))$ and intervals $(I_0 = [m_0^l;m_1^l],.., I_n = [m_{n}^l;m_{n+1}^l])$.
> > > If we don't use the contrast parameter ($\tau = 0$) then the hashed scalar weight values $\tilde w$ are defined as
> > >
> > > $$ \tilde w = \sum_{i=0}^n M_i^l \textbf{1}_{w \in  I_i}  $$
> > >
> > > Now let's assume that $\tau > 0$.
> > > The support of the weight values of this layer is $[\min\{w\in W^l\};\max\{w\in W^l\}]$, then the contrast parameter $\tau^l$ for layer $l$ gives us a threshold distance $s = (\max\{w\in W^l\}-\min\{w\in W^l\}) \times \tau$: local maxima whose distance is lower than this threshold are collapsed into a single one, *i.e.* the one that corresponds to the highest density mode in the weight distribution. We denote the index associated to this mode as
> > >
> > > $j_i = \text{argmax}$ \{ $d^l(M_j^l) |\forall j \in \{0,...,n\}\text{ s.t. } |M_j^l-M_i^l|\leq s $\}.
> > >
> > > Thus, the hashed weights are given by:
> > >
> > > $$\tilde w = \sum_{i=0}^n M_{j_i}^l \textbf{1}_{w \in I_i]}$$
> > >
> > > This corresponds to only keeping the local maxima that do not have another local maximum with a higher density within a distance defined as $\tau$\% of the weight value range for this layer.
> > >
> > > **Relation between $\tau$ and $|K^+_l|$:** By definition of $\tau$, the higher $\tau$ the lower the number of kept local maxima and the lower $|K^+_l|$. For limit values: if $\tau = 0$ then $|K^+_l|$ is simply the number of local maxima obtained by the kde method. If $\tau = 1$ then we keep only the local maximum associated with the highest density mode of the weight distribution, that is to say we have $|K^+_l| = 1$.
> > >
> > > **Choice of bandwidth:** we considered the mean and median (as well as multiples of these values) as natural candidates for an adaptive value for the bandwidth. We report different set-ups in the following table: the median provided the best trade-offs between accuracy preservation and compression rate, likely due to its robustness to outlier weight values.
> > >
> > > |            | unhashed model | 0.1$\times$median  | median | 10$\times$median | 0.1$\times$mean  | mean   | 10$\times$mean |
> > > | --- | --- | --- | --- | --- |  --- | --- | --- |
> > > | accuracy    | 93.46          | 93.46        | 93.41        | 92.95        | 93.46                | 93.40  | 93.02 |
> > > | compression | 0\%            | 94.0\%       | 99.0\%       | 99.3\%       | 92.8\%               | 97.6\% | 99.0\% |
> > >
> > >
> > > **Scalability:** As stated in the article, in Section 3 - l. 109-118, the hashing protocol is performed layer per layer independently. Thus, the suggested parallelization doesn't change its result, *i.e.* it doesn't cause any accuracy drop.
> > > As for the second suggestion (as expressed in our first response) that consists in sub-sampling the layer weight values to get identical density estimators much faster, we experimentally validated that it did not change the accuracy of the hashed function in practice: we achieved $99.9$\% compression for both ResNet 101 and 152 in 2 and 4.5 hours respectively (see our previous comment).

---

### Public Comment · ~Yuhang_Wu4 · 2022-08-27
**Request for releasing code**

I found you paper to be highly insightful. I was wondering if it is possible for you to release the source code of this paper?

---

### Decision · Program_Chairs · 2021-09-27

**Decision:**

Accept (Poster)

**Comment:**

I agree with the two of the reviewers that this contribution is novel and interesting. While the empirical evidence presented here could (and was) improved (in the rebuttals), I think that a sufficiently revised version of the paper (as suggested in the rebuttals) can be accepted for a poster at NeuriPS.